# Short-term persistence of foliar insecticides and fungicides in pumpkin plants and their pollinators

Jessie Lanterman Novotny[1], Keng-Lou James Hung[2], Andrew H. Lybbert[3], Ian Kaplan[4], Karen Goodell[5]*

1 Department of Biology, Hiram College, Hiram, Ohio, United States of America, 2 Oklahoma Biological Survey, University of Oklahoma, Norman, Oklahoma, United States of America, 3 Department of Biology, Scottsdale Community College, Scottsdale, Arizona, United States of America, 4 Department of Entomology, Purdue University, West Lafayette, Indiana, United States of America, 5 Department of Evolution, Ecology & Organismal Biology, The Ohio State University, Newark, Ohio, United States of America

* goodell.18@osu.edu

## Abstract

To minimize the risk to bees and other beneficial insects, plant protection chemicals are typically applied to pollinator-dependent crop plants when flowers are absent or unopened. However, this approach does not entirely remove the risk of pollinator exposure. Much research has focused on negative effects of systemic insecticides (e.g., seed treatments) on pollinators, but less is known about the level of hazard posed by translocation of non-systemic foliar-applied pesticides to pollen and nectar that bees consume. In this study we assess the frequency and persistence of six foliar-applied pesticides in pumpkin (*Cucurbita pepo*) tissues and in their bee visitors. We analyzed residues of three insecticides (carbaryl, lambda-cyhalothrin, permethrin) and three fungicides (chlorothalonil, quinoxyfen, triflumizole) in pumpkin leaves, pollen, and nectar collected from five farms in the north-central USA, one day before a spray event, and one, three, and seven days after. Bees foraging on pumpkin flowers were collected one day before and one day after spray and screened for the same pesticides. Overall, insecticides were present in 56% of leaf samples. Compared to leaves, fewer pollen (insecticide detected in 16%, fungicide in 16%) and nectar samples (14%, 0%) contained pesticides. We detected one insecticide (carbaryl) in two out of 69 samples of foraging bees, and only in male squash bees (not in bumble or honey bees), which have life history traits that bring them into prolonged close contact with the sprayed crop plants. The persistence of some agrochemicals in leaves, pollen, and nectar up to a week following application merits consideration when managing pollinator-dependent crops. Even pesticides that are traditionally considered contact-based and applied when flowers are unopened can reach pollen and nectar and produce measurable risk to bees.

**Data availability statement:** The raw data, as well as a data summary file and the R code we used to create the manuscript figures, are available through the online data repository Open Science Framework at: https://osf.io/mg5et/?view_only=527ab0b-982b4423192136a86dbbe3e70

**Funding:** This work was funded by United States Department of Agriculture through a National Institute of Food and Agriculture grant 2016-51181-25410 awarded to IK and KG. The funders had no role in study design, data collection and analysis, decision to publish, or preparation of the manuscript.

**Competing interests:** The authors have declared that no competing interests exist.

## Introduction

When applying chemical pesticides, growers must strike a balance between protecting crops and avoiding harm to non-target organisms like pollinating insects that contribute to crop yield [1,2]. As a result, much research has been done on the negative effects of agrochemicals on pollinators [3–5]. Recent research has focused on systemic insecticides applied to the soil or used as seed treatments, but less is known about the level and duration of hazard posed to pollinators by foliar-applied insecticides and fungicides [1,6–9]. In crops that are highly dependent on pollinators for fruit set, foliar sprays used in mid- to late-season to control diseases and insect pests are typically applied before the plants begin to bloom, or if flowers are present, late in the evening when bee activity is low and flower buds are closed [10]. That practice, while beneficial, does not entirely remove the risk of exposure for pollinators [1,11]. Pollinating insects may come into direct contact with recently sprayed foliage, soil, or contaminated dust particles, or consume contaminated pollen and nectar as adults or larvae [9,12–15]. We do not fully understand the translocation of foliar-applied chemicals from leaves and unopened flowers to the nectar and pollen upon which bees feed, and their persistence in or on plant tissues under field conditions (but see [15]). The dearth of data reflects the expense of pesticide residue analysis, as well as the difficulty of obtaining sufficient pollen and nectar directly from flowers [14,15].

In this study, we assess the frequency and persistence of foliar-applied insecticides and fungicides in leaves, pollen, and nectar of pumpkin (*Cucurbita pepo* L.), as well as in bees visiting pumpkin flowers. Pumpkins, like other cucurbit crops, have separate male and female flowers, so they rely on pollinating insects, especially bees, to set fruit. In the USA, pumpkin flowers are most commonly pollinated by several widely distributed, dietary-generalist eusocial bee taxa (western honey bee, *Apis mellifera* (L.)*,* and bumble bees, *Bombus* spp. (Latreille)) and by several solitary, *Cucurbita* pollen-specialist bees (*Xenoglossa* (*Peponapis*) *pruinosa* (Say) and other *Xenoglossa* spp., hereafter referred to collectively as "squash bees") [16,17]. *Cucurbita* specialists incur a higher risk from agrochemicals than do other bees. The females typically nest in soil directly beneath *Cucurbita* crop plants, and collect pollen exclusively from *Cucurbita* flowers, and the males spend much of their time sheltering in the wilted flowers [18–20]. For pollen-generalists like honey bees and bumble bees, *Cucurbita* nectar constitutes a smaller proportion of their diverse adult diets, and the larvae are also fed a mixture of pollen and nectar from many plant species. Although their generalist diets dilute the dose from any one source of contaminated pollen and nectar, eusocial bees in agricultural landscapes are subject to repeated long-term exposure to agrochemicals in their food and other materials they collect and store in the nest [8,21,22].

When pesticides are applied during *Cucurbita* bloom, the risk to its pollinators may be high. In *Cucurbita,* both staminate and pistillate flowers offer a large nectar reward, incentivizing bees to visit these flowers intensely during the several hours after sunrise when they are open [18]. Meanwhile, foliar- and soil-applied pesticides readily translocate to bee-collected pollen and nectar in cultivated *Cucurbita* [1,11,15,23–25]. There are currently >40 insecticides and >40 fungicides approved for use with *Cucurbita* crops in the USA [26], and growers rotate their use of different chemical classes to minimize pest resistance. Therefore, if agrochemicals persist in floral resources, soil, and other nesting materials, bees in agricultural fields can be exposed to a variety of chemicals simultaneously [12,22,27–29]. Even in cases where insecticide exposures are not acutely lethal, there have been numerous sublethal effects reported in pollinators, including decreased foraging efficiency [30–33], increased susceptibility to disease or parasites [21,34], reduced nest initiation [20,35], and reduced offspring production [20,36–38] or development [39]. Fungicides generally have

low toxicity to adult foraging bees [40], but can be toxic to larvae [41] and may disrupt bee digestion and immunity by altering gut microbiomes or histology [42–44]. A further complication is that certain agrochemicals have additive, or even interactive, negative effects on bees [45,46].

To determine the degree to which pollinators are exposed to foliar-applied agrochemicals that were sprayed while flowers were closed, we analyzed residues of three insecticides and three fungicides in pumpkin leaves, pollen, and nectar collected from five farms in the north-central USA, one day before a spray event as well as one, three, and seven days after the spray event. Specifically, we asked the following questions:

(1) How frequently are foliar insecticide and fungicide residues found in pumpkin pollen, nectar, and leaves in the week after spray?

(2) Does detection of residues in pumpkin tissues attenuate with time since spray?

(3) Are foliar insecticides and fungicides detected in bees visiting pumpkin flowers, and what level of hazard do they pose to pumpkin pollinators?

## Methods

### Study system

Our study took place on five pumpkin farms in central Ohio, USA, from 23 July – 16 August 2019 (S1 Table). Our sites were located within a predominantly agricultural area of temperate northcentral North America (mean maximum daily temperature during the study period = $29.4 \pm 1.7$ C, mean daily rainfall = $1.3 \pm 3.8$ mm, S2 Table). The farms were all conventional, commercial-scale working farms, with pesticide application decisions reflecting typical management regimes for our study region (ranging from prophylactic applications to integrated pest management). All growers used seeds treated with FarMore® FI400, which contains three fungicides (mefenoxam, fludioxonil, azoxystrobin) and the neonicotinoid insecticide thiamethoxam. Farmers grew multiple cultivars per field to appeal to their customers at their small local farm stands. We did not have information regarding the cultivar planted in each row and by peak flowering the pumpkin vines were often intertwined across rows. Foliar sprays were applied mid-season during flowering to manage insect pests like cucumber beetles (vectors of bacterial wilt), squash vine borers, aphids, and squash bugs [47] and fungal diseases like powdery and downy mildew, black rot, and Plectosporium blight [48] (S1 and S3 Tables). Growers indicated that they applied foliar sprays in the evening when pollinator activity was low and flower buds were unopened. Since previous studies have documented the translocation of systemic insecticides applied as a seed treatment to squash pollen and nectar [1,2,20], we chose to focus instead on the translocation of mid-season non-systemic foliar-applied pesticides to floral resources and bees. Application dates and pesticides used at each site are listed in S1 Table.

Pumpkin flowers in our system are visited by free-living bumble bees and squash bees [24]; nevertheless, four of the five farms also contracted or kept honey bees to supplement pollination. As in other cucurbit crops, pumpkin flowers are short-lived, with each flower opening around sunrise and lasting only several hours before wilting. The ephemeral nature of these flowers limits the routes through which non-systemic foliar-applied chemicals may enter pollen and nectar; the flowers we surveyed for residue analysis (see below) opened for the first time on the day we collected the pollen and nectar.

We received verbal permission from the land owners (who were also the growers) to access the field sites; no permits were required.

## Data collection

To determine the level of hazard to bees foraging in pumpkin fields after mid-season evening application of foliar sprays, we analyzed residues of six agricultural chemicals (three insecticides and three fungicides) in pumpkin leaves, pollen and nectar from flowers, and bee visitors to flowers. Growers decided independently what chemicals to apply and when. At each farm, we opportunistically assayed pumpkin tissues one day before, as well as one day, three days, and seven days after foliar spray events that involved one of our target agrochemicals (but not the day of the spray event), based on planned application schedules communicated by our collaborating growers. During each survey, we collected 20 pumpkin leaves as well as pollen and nectar from about 40 male flowers, from individual plants haphazardly selected across the entire field. We also collected up to 10 individuals of each of the following types of bees foraging on pumpkin flowers one day before and one day after spray events: bumble bee workers, honey bee workers, and male and female *X. pruinosa* squash bees. We prioritized bee collections on the day before and the day after application, when exposure to the pesticides was most likely. We decided not to collect additional bees on Day 3 and 7 after spray. Sampling at fewer time points allowed us to screen more samples on the day when contamination was most likely and to evaluate risk to three bee taxa, instead of honey bees only.

We collected the pumpkin leaves into resealable plastic bags. Bees were collected from flowers into individual microcentrifuge vials. We collected pollen and nectar by harvesting entire male flowers: anthers were collected into a 50-ml centrifuge tube, and nectar was extracted from the unopened nectar well (i.e., not depleted or contaminated by foraging bees) using a clean 50-µl micropipette tip glued onto the needle adapter of a 5-ml syringe. All samples were kept on ice and transferred to a -80°C freezer for storage.

Pesticide residue analysis was conducted on 1 g samples of leaf, pollen (half pollen and half anther tissue, hereafter referred to as 'pollen'), nectar, and bee tissue. We cut thin latitudinal strips from stacked pumpkin leaves at the widest part of the leaves near the center. These strips were chopped into small pieces using a sterile razor blade and from these pieces, a 1-g sample of tissue was weighed and placed in a 1.5-mL microcentrifuge tube. Pollen samples were scraped from the anthers with a sterile spatula to obtain a 0.5-g sample to which we added 0.5 g of tissue removed from the outer layer of 2 - 3 anthers due to the difficulty of obtaining 1 g of pure pollen. A 1-mL sample of nectar was measured into a clean 1.5-mL microcentrifuge tube. For the bee samples, it was necessary to combine 1 - 5 bees of the same species and sex collected from the same farm on the same day in order to make a 1-g sample (mean = 2.6 bees, SD = 1.4).

The tissue samples were kept at -80°C until shipped overnight on ice to a commercial agricultural chemical testing laboratory – SGS North America (Brookings, SD, USA). There, tissues were homogenized and residues extracted using the QuEChERS method (EN15662) [50], which has been commonly used to determine concentration of pesticide residues in leaf, pollen, and nectar tissue and has suitable recovery rates [15,51]. Chemical residues were analyzed using protocols outlined in the Association of Official Agricultural Chemists method [52]. Carbaryl and triflumizole residues were analyzed using liquid chromatography and tandem mass spectrometry (LC-MS/MS). Chlorothalonil, cyhalothrin (total), permethrin (cis and trans), and 5,7-dichloro-4-(p-fluorophenoxy) quinoline (i.e., quinoxyfen) were analyzed using gas chromatography mass spectrometry (GC-MS/MS) and concentrated from 2 mL to 1 mL. The concentrations of each residue were determined by comparing to standard curves including triphenyl phosphate (TPP) as the internal standard. Chemical concentrations were recorded in µg/ kg (parts per billion, hereafter 'PPB'). The lower limit of quantification for most chemicals was 10 µg/ kg, but was 20 µg/ kg for cyhalothrin and triflumizole.

## Data analysis

As growers decided independently what and when to spray (S1 and S3 Tables), the number of samples tested and detected was tallied separately for each chemical (and for each of two categories – insecticide and fungicide) using only the data from farms where each was used. We present the percent detection of each chemical in each tissue type over time (from 1 d before through 7 d after spray). All analyses were conducted using the R programming language, version 4.3.2 [53].

To assess risk to bees from contact with pumpkin leaves or consumption of pollen and nectar after foliar spray, three risk indices were calculated – Hazard Quotient ($HQ$), a variation of Hazard Quotient ($HQ_{PPB}$), and US EPA BeeREX Risk Quotient ($RQ$). First, Hazard Quotient ($HQ$) was calculated for each tissue type as the concentration of a chemical residue in the sample in µg/kg divided by $LD_{50}$ (the dose in µg/bee that would kill 50% of a test honey bee population) [54]. Acute oral $LD_{50}$ was used to calculate $HQ$ for pesticide concentrations in pollen and nectar, contact $LD_{50}$ for leaf, average of contact and oral $LD_{50}$ for bee samples. Honey bee $LD_{50}$ was used as $LD_{50}$ was not consistently available for bumble and squash bees (Table 1). Second, we created a variation of $HQ$ (hereafter referred to as $HQ_{PPB}$) that first converts $LD_{50}$ into the same units as the pesticide concentrations (µg/kg, or PPB, instead of µg/bee) to allow for direct comparisons between pesticide load and toxicity. $LD_{50}$ values were converted from µg/bee to µg/kg of bee tissue by dividing $LD_{50}$ values by 0.00012 kg (the average weight of a honey bee worker [55]). This index expresses pesticide concentration in the sample as the proportion of the $LD_{50}$ that a honey bee forager would have reached if it eats its own body weight in the contaminated nectar or pollen. The following is an example of how we calculated and interpreted $HQ_{PPB}$ versus $HQ$. The maximum concentration of carbaryl we detected in nectar 1 d after spray was 44.1 µg/kg. The oral $LD_{50}$ of carbaryl is 0.21 µg/bee (Table 1), or 1750 µg carbaryl per kg of nectar after conversion to PPB. Therefore, $HQ_{PPB}$ = 0.03, which indicates that concentration would not pose a high risk to bees because it represents only a small proportion of $LD_{50}$. In terms of risk to individual bees, if an adult honey bee foraging for pollen consumed 0.0000435 kg of pumpkin nectar the day after spray [56] at a concentration of 44.1 µg carbaryl/kg nectar, the bee would have ingested 0.002 µg of carbaryl, which is 1% of the $LD_{50}$ of 0.21 µg/bee. By comparison, for the same carbaryl concentration $HQ$ = 210, which

**Table 1. Summary of agrochemical sprays analyzed in pumpkin leaf, pollen, and nectar, and in bee visitors.**

| Chemical Residue | Category | Class | Commercial Formulation Used | Active Ingredient kg/L Concentrate | Contact Acute $LD_{50}$ Honey Bee | Oral Acute $LD_{50}$ Honey Bee | $logK_{ow}$ | $pK_a$ | Molecular Weight |
|---|---|---|---|---|---|---|---|---|---|
| Carbaryl | Insecticide | Carbamate | Sevin | 0.48 | 0.14 | >0.21 | 2.36 | 10.4 | 201.2 |
| Cyhalothrin, total | Insecticide | Pyrethroid | Province | 0.25 | 0.038 | 0.027 | 6.8 | 9 | 449.9 |
| Permethrin | Insecticide | Pyrethroid | Perm-Up | 0.38 | 0.024 | 0.13 | 6.1 | NA | 391.3 |
| Chlorothalonil | Fungicide | Organochlorine | Bravo | 0.50 | > 101 | >63 | 2.94 | neutral | 265.9 |
| Triflumizole | Fungicide | Imidazole | Procure | 0.48 | 20 | 14 | 4.77 | 3.7 | 345.8 |
| Quinoxyfen | Fungicide | Quinoline | Quintec | 0.25 | >100 | >100 | 5.1 | neutral | 308.1 |

$LD_{50}$ values (the amount of a chemical that is lethal to 50% of a test population, worst case of 24, 48, and 72 h values) are in µg/bee for contact and oral acute $LD_{50}$ in honey bees, *Apis mellifera*. Octanol-water partition coefficient ($logK_{ow}$) is the relative concentration of a chemical in n-octanol versus water at pH 7, 20°C. Higher values of $logK_{ow}$ indicate greater lipophilicity (and a lower affinity for water) and values above 4.5 are considered likely to adsorb and accumulate in lipid-rich tissues such as cuticular waxes or pollen. A chemical's ionizability is given as $pK_a$, the pH at which a chemical is 50% ionized, or in equilibrium between its undissociated and ionized state (calculated as the negative base-10 logarithm of the acid dissociation constant at 25°C). Chemicals with $pK_a$ < 7 are most likely to reach vascular tissue and mobilize systemically throughout the plant. A 'neutral' $pK_a$ indicates the chemical does not ionize under relevant plant conditions. Chemicals with molecular weights below 800 g mol⁻¹ are able to penetrate plant cuticles. Pesticide properties were obtained from the University of Hertfordshire Pesticide Properties Database [49].

would be interpreted as a potential risk to bees using a common level of concern (LOC) of $HQ > 50$ [57,58].

In addition to $HQ$ and $HQ_{PPB}$, we calculated contact and dietary *Risk Quotient* ($RQ_{contact}$, $RQ_{pollen}$, $RQ_{nectar}$) using the U.S. Environmental Protection Agency's BeeREX tool, which is intended for foliar sprays applied to crops in bloom [56]. Risk quotient has the advantage of taking into account the amount of the contaminated substance consumed or encountered by a typical honey bee forager [57]. The BeeREX tool calculates dietary $RQ$ separately for adult honey bee pollen and nectar foragers and compares empirical values of residue concentrations in pollen and nectar, multiplied by the typical amount of pollen and nectar consumed by a honey bee nectar or pollen forager, to oral acute $LD_{50}$. Contact $RQ$ is calculated by comparing the chemical application rate, multiplied by a constant that represents the typical amount of chemical encountered by a honey bee forager if it flies through a cloud of spray, to the contact acute $LD_{50}$. Ideally, to assess bee risk of pesticide contamination from leaf contact, this index should be further modified to also take into account the proportion of a bee's body that comes into contact with the plant when it is resting on foliage, but that data was not available. For further details on how the BeeREX tool calculates risk quotient, see the US EPA document Guidance for Assessing Pesticide Risks to Bees, Appendix 3 [59].

To interpret risk to bees associated with each tissue type and chemical, risk index values were compared to a pre-determined level of concern (LOC) for each index and interpreted as 'low risk' if below the LOC or 'low risk cannot be concluded' if exceeding LOC [57]. $HQ$ values above 50 [57,58] and $RQ$ values above 0.4 [59] are considered a potential risk to pollinators. A threshold LOC has not yet been established for $HQ_{PPB}$, but a simple and direct way to interpret it would be that values greater than 1 indicate a significant likelihood of acute bee toxicity (because if $HQ_{PPB} > 1$, the quantity of pesticide in a pollen or nectar sample that has the same mass as an average honey bee worker exceeds honey bee $LD_{50}$).

## Results

Foliar insecticide and fungicide spray residues were detected more frequently and in greater concentrations in pumpkin leaves than in pollen, nectar, or foraging bees (Figs 1, 2). Overall, more than half of leaf samples (56%) tested positive for at least one insecticide and 12% for at least one fungicide (Table 2). Based on risk assessment quotients (Table 3, S4 Table), insecticide concentrations in leaves often exceeded levels of concern (Fig 1). However, those indices assume that a foraging bee would actually come into contact with all the chemical present on or in the leaf sample. Compared to leaves, fewer pollen (insecticide detected in 16%, fungicide in 16%) and nectar samples (14%, 0%) contained pesticide residues and those that did typically had lower concentrations (Table 2). However, pollen contained relatively high levels of the insecticides carbaryl and permethrin after spray (Table 3). Of the six chemicals, only one insecticide (carbaryl) was detected in nectar after spray. Similarly, we had low detection of insecticides (3%) and no fungicides in adult foraging bees (Table 2). Carbaryl was present in two bee samples at levels below acute $LD_{50}$; both of which were male *X. pruinosa* that were collected one day after a spraying event.

Among the six agrochemicals tested, only carbaryl, the most commonly used insecticide, was detected in all four tissue types – leaves, pollen, nectar, and bees (Table 2). No additional chemicals were found in nectar or bee samples. The insecticides lambda cyhalothrin and permethrin were both detected in leaves, but only the latter was present in pollen as well. Two of three fungicides (triflumizole and quinoxyfen) were detected in pumpkin tissues – quinoxyfen in pollen only, and triflumizole in both leaves and pollen. The fungicide chlorothalonil was not detected in any pumpkin tissues (similar to the results of Bloom et al. [24]), even though

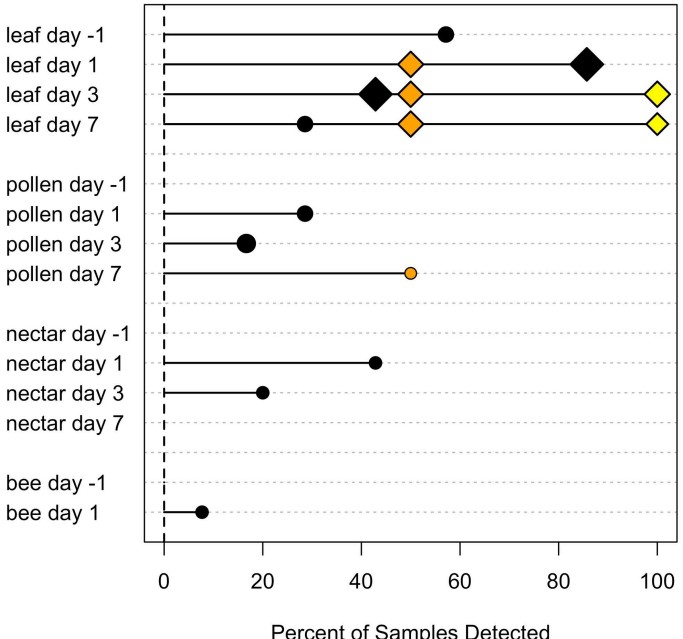

**Fig 1. Insecticide detection in pumpkin tissues and bee visitors.** The percent of samples in which a residue was detected is shown for each tissue type and time since spray. Percent detection of each chemical was calculated separately using only the data from farms where each was applied. Chemicals are color coded as follows: carbaryl (black), permethrin (orange), lambda cyhalothrin (yellow). Point size corresponds to the following six categories of maximum chemical concentration: < 100 PPB, 100 – 500, 501 – 1000, 1001 – 5000, 5001 – 10000, >10000. A diamond shape indicates that at the maximum concentration detected, $HQ_{PPB}$ exceeded the level of concern for bee risk assessment ($HQ_{PPB} > 1$). A circle indicates $HQ_{PPB} < 1$.

it was used by three out of five of our growers. The maximum concentration varied strongly among chemicals (Table 2).

Typically, either the proportion of contaminated samples or the maximum concentration of insecticides in pumpkin tissues decreased over the week following foliar application (Fig 1). Overall, at least one insecticide was found in 75% of leaves the day after spray, and 50% on Days 3 and 7 (Table 4). The likelihood of detecting insecticides in pollen the day after spray was 29%, and decreased to 13% by Day 7. The day after spray 43% of nectar samples tested positive for the insecticide carbaryl, which decreased to 0% by Day 7. Although detection decreased over time, our data suggest that some insecticides persisted in leaves and pollen beyond one week. For example, carbaryl was found in 4 out of 7 leaf samples the day before spray (Table 5) across three farms, including one positive sample from a farm where that chemical was not applied during our study period. At one farm that applied carbaryl, then applied it again nine days later at a lower rate, carbaryl was detected in leaves during both sampling rounds except for Day 7 after the second application. It was not common among growers in this study to apply multiple insecticides at a time, but at one farm that did use two in the same spray event, 50% of leaves the next day (but not pollen, nectar, or bees) contained both chemicals (Supporting Information, S5 Table).

The percent of pumpkin tissue samples that contained fungicides remained relatively constant in the week following application (Table 4), though the maximum concentrations in leaves decreased with time (Fig 2). One fungicide (quinoxyfen) was also detected in 1 of 4 pollen samples the day before spray (Supporting Information, S9 Table). When growers applied more than one fungicide over the course of the study period, multiple were taken up

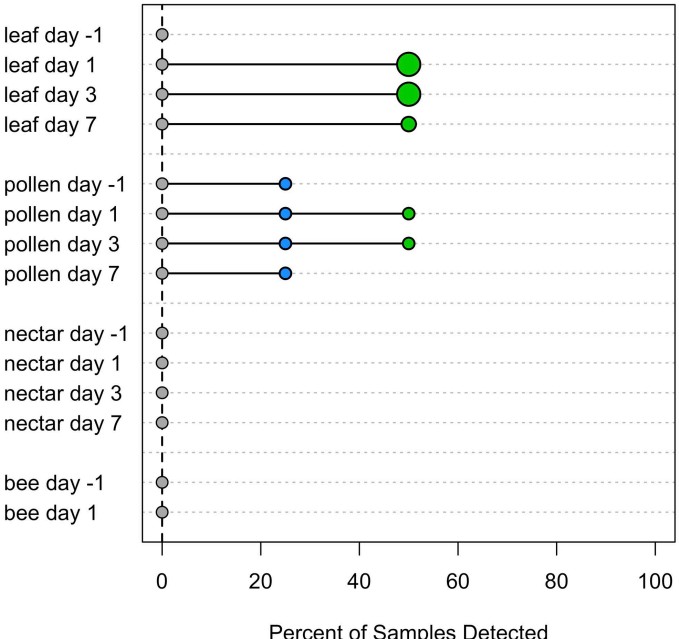

**Fig 2. Fungicide detection in pumpkin tissues and bee.** The percent of samples in which a residue was detected is shown for each tissue type and time (days since spraying event). Percent detection of each chemical was calculated separately using only the data from farms where each was applied. Chemicals are color coded as follows: triflumizole (green), quinoxyfen (blue), chlorothalonil (gray). Point size and shape are as in Fig 1.

by pollen, but not by other tissues (Supporting Information, S5 Table). For example, at one farm that applied quinoxyfen and triflumizole fungicides a week apart, both were detected in 2 out of 6 pollen samples after the second spray event.

## Discussion

Much research has already reported acute and sublethal effects of systemic insecticides on beneficial insects like pollinators in agricultural landscapes [6,8,9]. There have been fewer empirical studies on translocation of non-systemic pesticides sprayed on crop foliage to non-target plant regions (but see [60]), but models predict these chemicals can move into floral tissues under certain conditions based on their chemical properties [61]. Here we add to the emerging knowledge of frequency and persistence of non-systemic foliar-applied pesticides in crop leaves, pollen, nectar, and pollinating bees. As expected, we detected insecticide residues with high frequency and concentration in pumpkin leaves through the week following application in concentrations that may pose a risk to bees if they have extended contact with leaves (e.g., while females groom pollen onto their scopa or rest under foliage during inclement weather). On day 1 after spray, insecticides and fungicides were present in floral tissue (especially pollen), which raises questions about whether spray droplets leak into unopened flowers at the time of application or translocate later through foliage or petals and become incorporated into nectar and pollen. We detected insecticide residues in few bee samples the day after spray, and the bees that tested positive (male squash bees) have life history traits that bring them into prolonged contact with sprayed crop plants.

A key strength of our study is that it was a 'natural' experiment conducted in fields managed by local growers using chemicals and rates of application that they typically use. Tradeoffs of this approach are an unbalanced design, lack of control over the crop variety

**Table 2. Summary of detection of six agrochemical residues in pumpkin tissues and in bee visitors to flowers from one day before to one week after foliar spray application.**

| Chemical | Leaf | | | | Pollen | | | | Nectar | | | | Bee | | | |
|---|---|---|---|---|---|---|---|---|---|---|---|---|---|---|---|---|
| | N | $N_{Pos.}$ | Min | Max | N | $N_{Pos.}$ | Min | Max | N | $N_{Pos.}$ | Min | Max | N | $N_{Pos.}$ | Min | Max |
| Carbaryl | 28 | 15 | **12.4** | **27980.0** | 27 | 3 | **26.7** | **583.1** | 24 | 4 | **11.0** | **44.1** | 69 | 2 | **22.8** | **89.1** |
| Permethrin | 8 | 3 | **2071.7** | **4755.5** | 7 | 1 | **57.9** | **57.9** | 7 | 0 | – | – | 44 | 0 | – | – |
| L. Cyhalothrin | 4 | 2 | **677.6** | **1362.9** | 4 | 0 | – | – | 4 | 0 | – | – | 0 | – | – | – |
| **Insecticide** | 32 | 18 | – | – | 32 | 5 | – | – | 28 | 4 | – | – | 69 | 2 | – | – |
| Triflumizole | 8 | 3 | 305.4 | 9023.1 | 8 | 2 | 23.3 | 24.7 | 6 | 0 | – | – | 0 | – | – | – |
| Quinoxyfen | 0 | – | – | – | 17 | 4 | 26.8 | 37.1 | 13 | 0 | – | – | 15 | 0 | – | – |
| Chlorothalonil | 16 | 0 | – | – | 16 | 0 | – | – | 15 | 0 | – | – | 25 | 0 | – | – |
| **Fungicide** | 24 | 3 | – | – | 25 | 4 | – | – | 21 | 0 | – | – | 25 | 0 | – | – |

Chemicals are ordered from most to least frequently detected within the two categories – insecticide and fungicide. The total number of samples tested ($N$) and the number in which a chemical residue was detected (number positive, $N_{Pos.}$) is given for each tissue. The number tested and detected was tallied separately for each chemical (and for each category) using only the data from farms where that chemical was applied. Minimum and maximum chemical concentrations are given in µg/ kg (PPB). Bold font indicates that a concentration exceeded level of concern (LOC) for at least one of our three risk assessment indices ($HQ > 50$, $HQ_{PPB} > 1$, $RQ > 0.4$); bold underlined font indicates it exceeded LOC for all three. The number of farms where each chemical was applied, the rate of application, and the total number of spray events is provided in S1 Table.

**Table 3. Risk assessment for bees visiting pumpkin flowers within one week after foliar spray application, based on the maximum chemical concentrations found in each tissue.**

| Tissue | Chemical | HQ | HQPPB | RQ |
|--------|----------|-----|-------|-----|
| Leaf | Carbaryl | **199857.14** | **23.98** | **96.06** |
| | Permethrin | **198145.83** | **23.78** | **67.19** |
| | Cyhalothrin | **35865.79** | **4.30** | **14.14** |
| | Triflumizole | **451.16** | 0.05 | 0.05 |
| | Chlorothalonil | ND | ND | 0.07 |
| Pollen | Carbaryl | **2776.67** | 0.33 | 0.01 |
| | Permethrin | **445.38** | 0.05 | <0.01 |
| | Cyhalothrin | ND | ND | 0.00 |
| | Triflumizole | 1.76 | 0.00 | <0.01 |
| | Quinoxyfen | 0.37 | <0.01 | <0.01 |
| | Chlorothalonil | ND | ND | 0.00 |
| Nectar | Carbaryl | **210.00** | 0.03 | 0.06 |
| | Permethrin | ND | ND | <0.01 |
| | Cyhalothrin | ND | ND | 0.00 |
| | Triflumizole | ND | ND | <0.01 |
| | Quinoxyfen | ND | ND | <0.01 |
| | Chlorothalonil | ND | ND | 0.00 |
| Bee | Carbaryl | **509.14** | 0.06 | NA |
| | Permethrin | ND | ND | NA |
| | Quinoxyfen | ND | ND | NA |
| | Chlorothalonil | ND | ND | NA |

Three risk quotients were calculated for each tissue: $HQ$, $HQ_{PPB}$, and $RQ$. $RQ_{pollen}$ and $RQ_{nectar}$ estimate risk for a honey bee foraging on pollen or nectar, respectively, based on our empirical values of chemical concentration in pollen and nectar. $RQ_{leaf}$ (or contact $RQ$) was based on application rate of the pesticide and not on empirical concentration in the samples. Therefore, a value for chlorothalonil leaf $RQ$ is given even though it was not detected in our leaf samples. Bold font indicates the value exceeded LOC ($HQ > 50$, $HQ_{PPB} > 1$, $RQ > 0.4$) and, therefore, low risk cannot be concluded [57]. Only pesticides for which data were available were included in this table. ND indicates that a tissue was tested for the chemical, but not detected. For risk assessment based on minimum chemical concentrations, see Supporting Information S4 Table.

(most plantings were mixed cultivars), and low sample sizes for some agrochemicals and time points, which limited our ability to draw broader conclusions about their mobility in plants and the hazard they pose to pollinators.

## Detection of foliar-applied pesticides in pumpkin tissues and pollinators in the week after spray

There are two likely explanations for how non-systemic pesticides sprayed on foliage still reached pumpkin pollen, nectar, and foraging male squash bees, even though flowers were unopened during application. First, chemical-laden droplets may have leaked into flowers during spray and directly contaminated pollen and nectaries if the buds were not tightly sealed. Similarly, spray droplets may have also accumulated in wilted flowers where male squash bees were resting for the night, although only one chemical out of six (carbaryl) was detected in squash bees. Second, chemicals on the leaf surface or in stomal apertures may have crossed lipid boundaries, mobilized in phloem, and become incorporated into developing flowers. Environmental conditions, the addition of adjuvants (*e.g.*, surfactants), and

**Table 4. Detection of insecticide and fungicide residues in pumpkin tissues and bees by days since foliar spray application.**

| Tissue | Time | Any Insecticide(s) | Any Fungicide(s) |
|---|---|---|---|
| Leaf | -1 | 50 | 0 |
| | 1 | 75 | 17 |
| | 3 | 50 | 17 |
| | 7 | 50 | 17 |
| | total | 56 | 13 |
| Pollen | -1 | 0 | 14 |
| | 1 | 29 | 20 |
| | 3 | 14 | 17 |
| | 7 | 13 | 17 |
| | total | 16 | 16 |
| Nectar | -1 | 0 | 0 |
| | 1 | 43 | 0 |
| | 3 | 17 | 0 |
| | 7 | 0 | 0 |
| | total | 14 | 0 |
| Bee | -1 | 0 | 0 |
| | 1 | 8 | 0 |
| | total | 3 | 0 |

The percent of samples (rounded to the nearest integer) in which any one or more insecticides or fungicides were detected is given by tissue type and time since spray (-1 = one day before, 1 = one day after, 3 = three days after, 7 = seven days after). Percent detection was calculated separately for insecticides and fungicides using only data from farms where sprays from each category were used.

**Table 5. Detection of carbaryl insecticide in pumpkin and bee tissues before a spray event through one week after.**

| Tissue | Time | N | $N_{Pos.}$ | Min | Max | HQ | $HQ_{PPB}$ | RQ |
|---|---|---|---|---|---|---|---|---|
| Leaf | -1 | 7 | 4 | 19.0 | 225.2 | **1608.5** | 0.19 | NA |
| | 1 | 7 | 6 | 13.8 | 27980.0 | **199857.1** | **23.98** | NA |
| | 3 | 7 | 3 | 12.4 | 19020.0 | **135857.1** | **16.30** | NA |
| | 7 | 7 | 2 | 145.3 | 244.9 | **1749.3** | 0.21 | NA |
| Pollen | -1 | 7 | 0 | – | – | 0.00 | 0.00 | 0.00 |
| | 1 | 7 | 2 | 26.7 | 108.8 | **518.1** | 0.06 | 0.01 |
| | 3 | 6 | 1 | 583.1 | 583.1 | **2776.6** | 0.33 | <0.01 |
| | 7 | 7 | 0 | – | – | 0.00 | 0.00 | 0.00 |
| Nectar | -1 | 7 | 0 | – | – | 0.00 | 0.00 | 0.00 |
| | 1 | 7 | 3 | 14.4 | 44.1 | **210.1** | 0.03 | 0.06 |
| | 3 | 5 | 1 | 11.0 | 11.0 | **52.4** | 0.01 | 0.02 |
| | 7 | 5 | 0 | – | – | 0.00 | 0.00 | 0.00 |
| Bee | -1 | 43 | 0 | – | – | 0.00 | 0.00 | NA |
| | 1 | 26 | 2 | 22.8 | 89.1 | **509.2** | 0.06 | NA |

Time since spray is indicated as -1 = one day before, 1 = one day after, 3 = three days after, 7 = seven days after. The number of samples tested ($N$) and the number in which carbaryl was detected (**$NPos.$**) is given for each tissue and time. Minimum and maximum concentrations are in units of μg/ kg (PPB). Risk quotients $HQ$, $HQ_{PPB}$, and $RQ_{pollen}$ and $RQ_{nectar}$ are based on the maximum residue concentration. $RQ_{leaf}$ is not available by day because it was estimated from initial application rate, not leaf concentration. Bold font indicates concentration exceeded a threshold level of concern of $HQ > 50$, $HQ_{PPB} > 1$, or $RQ > 0.40$. For detection of the other chemicals by day and tissue type, see Supporting Information S6–S9 Tables.

the chemical properties of pesticides such as water solubility, lipophilicity, ionizability, and molecular weight all contribute to their ability to infiltrate a plant's epidermis and translocate to non-target regions [62–66]. These xenobiotic chemicals move through plants along both lipophilic and aqueous pathways, and undergo complex and transformative interactions [65]. Pesticides applied to crop plants are typically slightly lipophilic so that they can permeate the cuticle, but not be completely adsorbed in intra-cuticular non-crystalline wax [64]. They are also often weak electrolytes that can dissolve in water and be transported to different plant compartments [62,63]. Overall, pesticides that are somewhat lipophilic weak acids are most mobile; they are bioactive under certain conditions yet do not ionize at the pH levels of aqueous reservoirs inside plant cells and become 'ion trapped' before reaching phloem to disperse [62].

The lipophilicity and other properties of our target agrochemicals can partially explain their biological fate in pumpkin tissues in our dataset [62–65]. For example, three of the four pesticides we detected in pollen – permethrin, quinoxyfen, and triflumizole – are lipophilic ($logK_{OW} > 4.5$, Table 1), making them likely to adsorb onto lipid-rich tissues like the pollen coat, if they are not first trapped in leaf cuticle waxes. Another insecticide we detected in pollen – carbaryl – has only moderate lipophilicity ($logK_{OW} = 2.4$), giving it more potential to mobilize vascularly and be incorporated into developing floral tissue. Consistent with this reasoning, we recorded a five-fold increase in carbaryl concentrations in pollen from the first to the third day after spray (Table 5). However, our low sample sizes make us unable to resolve whether those values are within the typical range of variation for pollen or are evidence of systemic uptake. Another consideration for carbaryl is that it also has a low molecular weight and is a very weak acid (dissociates more readily at low pH levels), making it likely to cross membranes and to ionize and bind with compounds in relatively acidic compartments inside cells like vacuoles (pH ~ 5) before it reaches phloem (pH ~ 8) [62]. These properties contribute to its persistence in leaves, instead of translocation to pollen and nectar that bees eat. This example of carbaryl demonstrates the complexity of interactions pesticides undergo within crop plants; no one factor can fully explain their presence.

In addition to the properties of the agrochemicals, differences in the composition of plant tissues and the timing of their availability also influence their uptake of pesticides. For example, pollen has a higher lipid and protein content [67] and takes more time to develop than nectar, which may contribute to its greater pesticide load in ours and other studies [1,11,15,19]. We detected four out of six agrochemicals (two insecticides and two fungicides) in pollen after foliar application. Fungicide was detected almost as frequently in pollen as in leaves (Table 4). In contrast, only one insecticide and no fungicides were found in nectar. Nectar is secreted after flower anthesis [68], and so should not have been present in unopened flowers during spray.

Based on our detection of pesticides in pollen, it is likely that bee larvae fed on pumpkin pollen from these fields were exposed to multiple agrochemicals simultaneously, with the potential for additive, or even interactive toxicity [41,45]. However, in this field study, wild bee nest locations were unknown and we were not able to test larvae from the nests of the adult foragers we collected. Laboratory trials with contaminated pollen fed to larvae of honey bees or other agriculturally important pollinators could illuminate the effects of consuming pesticide residues on larval development and survival [39,41,69,70]. Population-level studies that investigate multiple exposure routes and outcomes for larvae and adult bees such as that by Willis-Chan and Raine [20] are best able to assess total risk.

Consumption of contaminated pollen and nectar is arguably a greater concern for bees than leaf contact exposure because it can impact larvae as well as adults and lead to population-level effects [14,15]. However, the persistence of the insecticide carbaryl in

leaves between spraying events also raises concern from the pollinator perspective because it prolongs their risk of exposure. In addition, the high concentrations of pesticides in leaves during the week after foliar spray led to the highest bee risk quotient values in our dataset (Table 3). However, risk assessments that aim to quantify bee toxicity from leaf contact may overestimate the true risk to bees since they assume that a bee actually encountered the entire dose of the chemical present in the leaf sample. Instead, only the bee's lower legs and tip of the abdomen contact a leaf on which it is resting (*authors' personal observations*).

Bee contamination was infrequent in our dataset. We detected one insecticide (carbaryl, the most widely used by growers in this study) in two out of 69 total bee samples and at concentrations that were not acutely lethal. Both positive bee samples were from the same field on the same date and consisted of bees that we expected to have the highest risk of exposure – males of *X. pruinosa*, a *Cucurbita* pollen specialist bee species, which spend much of their adult lives resting in wilted squash flowers [1]. On that particular farm, carbaryl was also present in one out of three nectar samples the day after spray. Therefore, we are unable to distinguish whether the male squash bees were exposed directly while resting in wilted flowers at the time of spray, or via translocation of the pesticide through the plant to the nectar. Wild bees foraging or nesting in pumpkin fields are also at risk of exposure to agrochemicals through non-dietary media (*e.g.,* contaminated soils or water in or near crop fields), which can contribute to multiple additive exposures, but these routes were beyond the scope of our study [13,19,20,71].

Since so few bees tested positive the day after spray application (when they would have come into contact with the greatest concentrations of pesticides), it seems unlikely that additional sampling on Days 3 and 7 would have revealed higher contamination rates. However, our data may be subject to survivorship bias; our sampling method only captured living bees that were still able to forage the day after a spray event, and not those that had encountered or accumulated a lethal dose. Monitoring nearby honey bee hives for worker losses before and after pesticide applications could have offered further insight [72].

## Conclusions

Our results strengthen the evidence that agricultural chemicals applied to foliage are present in crop plant pollen and nectar, as well as leaves after spray, at least in the short term. Therefore, applying pesticides to crops when flowers are present, but unopened, fails to completely protect pollinators from exposure. The six chemicals we tested differed in their patterns of detection in pumpkin tissues, likely due to their physicochemical properties. Pesticides were infrequently found in adult bees foraging on pumpkin flowers in our dataset, and only in squash bees and not in social taxa, likely because of bee natural history traits. The persistence of some pesticides in leaves and pollen up to a week following application, often at concentrations that exceeded levels of concern for bees, should be a consideration when managing pollinator-dependent crops. If standardized translocation rates for crop plant pollen and nectar were included in pesticide properties databases, it would allow for better integrated pest management decisions in pollinator-dependent crops. More research is also needed on the effects of consumption of contaminated pollen by bee larvae and how that translates to population-level effects.

## Supporting Information

**S1 Table. Summary of pesticides applied during the study period that we tested for in pumpkin and bee tissues.** To preserve the anonymity of growers, farms are listed by identifiers A – E. The approximate location of each grower's farm, rounded to the nearest 1/10th

decimal degree for anonymity, is as follows: A – 40.1N, -82.8W; B – 40.0, -82.4; C – 40.1, -82.7; D – 40.0, -82.6; E – 39.9, -82.8. Spray events during the study period are numbered sequentially within farm. For example, at farm B spray event 1, the grower applied both carbaryl and permethrin on the same day. Pesticide residue names are given as their commercial formulations and % active ingredient(s). The rate of application is given in fluid ounces per hectare, either as a range or a single concentration, as reported by the growers.
(PDF)

**S2 Table. Daily weather records during the study period from 23 July – 16 August, 2019, from the airport nearest our sites (John Glenn Columbus International Airport).**
(PDF)

**S3 Table. Additional fungicides applied during the study period that we were not able to test for.** Farms and spray dates are as listed in S1 Table. Pesticide residue names are given as their commercial formulations and % active ingredient(s). The rate of application is given in fluid ounces per hectare, either as a range or a single concentration, as reported by the growers.
(PDF)

**S4 Table. Risk assessment for bees visiting pumpkin within one week after foliar spray, based on the minimum chemical concentrations detected in each tissue type.** Three risk quotients were calculated for each tissue: $HQ$ (Hazard Quotient), $HQ_{PPB}$ (Hazard Quotient in which LD50 was first converted to bee-relevant PPB), and $RQ$ (Risk Quotient from the US EPA BeeREX tool). See Methods for details of index calculations. Bold font indicates the index value exceeded the threshold level of concern ($HQ > 50$, $HQ_{PPB} > 1$, $RQ > 0.40$), and, therefore, low risk cannot be concluded. Only pesticides for which data were available were included in this table. For risk assessment based on the maximum chemical concentrations found in each tissue, see main text Table 3.
(PDF)

**S5 Table. Detection of one versus multiple insecticide and fungicide residues in pumpkin and bee tissues at farms that applied multiple chemicals.** The percent of samples (rounded to the near integer) in which any one or two different chemicals were detected was calculated separately for insecticides and fungicides using only data from farms where multiple sprays from each category were used. Four out of five farms used a mix of both insecticides and fungicides during the study period, but no farms sprayed all three of our focal insecticides or all three fungicides during the study. Time since foliar application is indicated as -1 = one day before, 1 = one day after, 3 = three days after, 7 = seven days after.
(PDF)

**S6 Table. Detection of permethrin insecticide in pumpkin and bee tissues before a spray event through one week after.** Time since spray is indicated as -1 = one day before, 1 = one day after, 3 = three days after, 7 = seven days after. The number of samples tested (*N*) and the number in which permethrin was detected (**NPos.**) is given for each time. Minimum and maximum concentrations are in units of μg/ kg (PPB). Risk quotients $HQ$, $HQ_{PPB}$, and $RQ_{pollen}$ and $RQ_{nectar}$ are based on the maximum residue concentration. $RQ_{leaf}$ is not available by day because it was estimated from initial application rate, not leaf concentration. Bold font indicates concentration exceeded a threshold level of concern of $HQ > 50$, $HQ_{PPB} > 1$, or $RQ > 0.40$.
(PDF)

**S7 Table. Detection of lambda cyhalothrin insecticide in pumpkin and bee tissues before a spray event through one week after.** Time since spray is indicated as -1 = one day before,

1 = one day after, 3 = three days after, 7 = seven days after. The number of samples tested (*N*) and the number in which lambda cyhalothrin was detected (**NPos.**) is given for each time. Minimum and maximum concentrations are in units of μg/ kg (PPB). Risk quotients *HQ*, $HQ_{PPB}$, and $RQ_{pollen}$ *and* $RQ_{nectar}$ are based on the maximum residue concentration. $RQ_{leaf}$ is not available by day because it was estimated from initial application rate, not leaf concentration. Bold font indicates concentration exceeded a threshold level of concern of *HQ* > 50, $HQ_{PPB}$ > 1, or RQ > 0.40.
(PDF)

**S8 Table.  Detection of triflumizole fungicide in pumpkin and bee tissues before a spray event through one week after.** Time since spray is indicated as -1 = one day before, 1 = one day after, 3 = three days after, 7 = seven days after. The number of samples tested (*N*) and the number in which triflumizole was detected (**NPos.**) is given for each time. Minimum and maximum concentrations are in units of μg/ kg (PPB). Risk quotients *HQ*, $HQ_{PPB}$, and $RQ_{pollen}$ *and* $RQ_{nectar}$ are based on the maximum residue concentration. $RQ_{leaf}$ is not available by day because it was estimated from initial application rate, not leaf concentration. Bold font indicates concentration exceeded a threshold level of concern of *HQ* > 50, $HQ_{PPB}$ > 1, or RQ > 0.40.
(PDF)

**S9 Table.  Detection of quinoxyfen fungicide in pumpkin and bee tissues before a spray event through one week after.** Time since spray is indicated as -1 = one day before, 1 = one day after, 3 = three days after, 7 = seven days after. The number of samples tested (*N*) and the number in which quinoxyfen was detected (**NPos.**) is given for each time. Minimum and maximum concentrations are in units of μg/ kg (PPB). Risk quotients *HQ*, $HQ_{PPB}$, and $RQ_{pollen}$ *and* $RQ_{nectar}$ are based on the maximum residue concentration. $RQ_{leaf}$ is not available by day because it was estimated from initial application rate, not leaf concentration. Bold font indicates concentration exceeded a threshold level of concern of *HQ* > 50, $HQ_{PPB}$ > 1, or RQ > 0.40.
(PDF)

## Acknowledgements

We are grateful for the cooperation of our participating pumpkin growers. We thank Jack Omori for help collecting field data and processing samples in the lab. We also thank three anonymous reviewers for suggestions that improved our manuscript.

## Author contributions

**Conceptualization:** Jessie Lanterman Novotny, Keng-Lou James Hung, Karen Goodell.

**Data curation:** Jessie Lanterman Novotny, Keng-Lou James Hung, Andrew H. Lybbert, Karen Goodell.

**Formal analysis:** Jessie Lanterman Novotny, Keng-Lou James Hung.

**Funding acquisition:** Ian Kaplan, Karen Goodell.

**Investigation:** Andrew H. Lybbert, Karen Goodell.

**Methodology:** Jessie Lanterman Novotny, Keng-Lou James Hung, Andrew H. Lybbert, Ian Kaplan, Karen Goodell.

**Project administration:** Karen Goodell.

**Resources:** Karen Goodell.

**Supervision:** Karen Goodell.

**Validation:** Jessie Lanterman Novotny.

**Visualization:** Jessie Lanterman Novotny.

**Writing – original draft:** Jessie Lanterman Novotny, Keng-Lou James Hung.

**Writing – review & editing:** Jessie Lanterman Novotny, Keng-Lou James Hung, Andrew H. Lybbert, Ian Kaplan, Karen Goodell.

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
