## [Decision Letter · Decision Letter 0]

5 Nov 2024

PONE-D-24-42091Short-term persistence of foliar insecticides and fungicides in pumpkin plants and their pollinatorsPLOS ONE

Dear Dr. Goodell,

Thank you for submitting your manuscript to PLOS ONE. After careful consideration, we feel that it has merit but does not fully meet PLOS ONE’s publication criteria as it currently stands. Therefore, we invite you to submit a revised version of the manuscript that addresses the points raised during the review process.

We look forward to receiving your revised manuscript.

Kind regards,

Nicolas Desneux

Academic Editor

PLOS ONE

Journal Requirements:

1. When submitting your revision, we need you to address these additional requirements. Please ensure that your manuscript meets PLOS ONE's style requirements, including those for file naming. The PLOS ONE style templates can be found at https://journals.plos.org/plosone/s/file?id=wjVg/PLOSOne_formatting_sample_main_body.pdf and https://journals.plos.org/plosone/s/file?id=ba62/PLOSOne_formatting_sample_title_authors_affiliations.pdf 2. Please note that PLOS ONE has specific guidelines on code sharing for submissions in which author-generated code underpins the findings in the manuscript. In these cases, we expect all author-generated code to be made available without restrictions upon publication of the work. Please review our guidelines at https://journals.plos.org/plosone/s/materials-and-software-sharing#loc-sharing-code and ensure that your code is shared in a way that follows best practice and facilitates reproducibility and reuse. 3. In your Methods section, please provide additional information regarding the permits you obtained for the work. Please ensure you have included the full name of the authority that approved the field site access and, if no permits were required, a brief statement explaining why. 4. Thank you for stating the following financial disclosure: [This work was funded by United States Department of Agriculture through a National Institute of Food and Agriculture grant 2016-51181-25410 awarded to IK and KG.]. Please state what role the funders took in the study.  If the funders had no role, please state: "The funders had no role in study design, data collection and analysis, decision to publish, or preparation of the manuscript." If this statement is not correct you must amend it as needed. Please include this amended Role of Funder statement in your cover letter; we will change the online submission form on your behalf.

Reviewers' comments:

Reviewer's Responses to Questions

**Comments to the Author**

1. Is the manuscript technically sound, and do the data support the conclusions?

Reviewer #1: Yes

Reviewer #2: Yes

Reviewer #3: Yes

2. Has the statistical analysis been performed appropriately and rigorously? 

Reviewer #1: Yes

Reviewer #2: Yes

Reviewer #3: Yes

3. Have the authors made all data underlying the findings in their manuscript fully available?

Reviewer #1: Yes

Reviewer #2: Yes

Reviewer #3: No

4. Is the manuscript presented in an intelligible fashion and written in standard English?

Reviewer #1: Yes

Reviewer #2: Yes

Reviewer #3: Yes

5. Review Comments to the Author

Reviewer #1: The present manuscript was well written, its objectives and the rationale of the study were clearly stated, the methods were adequate. Experiments were conducted rigorously, with appropriate controls. The conclusions were drawn appropriately based on the data presented. However, some suggestions and comments were provided in the attached copy of the submitted text.

Reviewer #2: The present work contains important information regarding the exposure of potential pollinators to pesticides, considering the application of these products when the flowers are no longer secreting nectar and/or have pollen available.

The strength of the article is the fact that the authors collected data in the field, in a real situation of pesticide use and potential exposure of non-target insects. The weakness is related to the low number of repetitions, especially for the data obtained with the bees.

Here are some notes that I think the authors should pay more attention to:

Despite the importance of the information presented in the introduction section, this section is excessively long. On the other hand, more information about the translocation of pesticides in plants should be included in the text. Since it may be difficult to include this information in the introduction of the article, I believe it would be better presented in the discussion section.

In the materials and methods section, the bee species should be described rather than just mentioned generically.

Regarding the materials and methods section, the description of the sample preparation for the pesticide quantification analyses could be better detailed.

The names of the pumpkin cultivars should be mentioned in the text, as well as the general growing conditions. Even more importantly, the environmental conditions should be specified: include data on temperature, sunlight, and precipitation.

What is the size of the cultivation areas? Could the bees have been contaminated in adjacent areas?

Some researchers have reported difficulties in detecting/quantifying pesticides in pollen samples. Is it possible to assert that the quantities observed in the pollen samples were not underestimated? I believe that a more detailed description of the detection method could better support the technique chosen for observing pesticides in pollen.

If a larger number of bees had been collected, regression analyses could have been established to understand the exposure time of these insects after spraying.

Table 1 is not mentioned in the text, nor are units provided for LD50.

The description of the results is very direct, making it difficult to understand. A more didactic and clear presentation could enhance comprehension, especially in the initial part.

The data on the pesticide application mixtures should be included in the work: what doses/dilutions were used? Which pest was targeted? What is the concentration of the active ingredients in the spraying mixtures? Were the producers accurate in preparing the pesticide mixtures?

Reviewer #3: To determine how non-systemic pesticides, persist and translocate on crops and whether their application to crops with closed or absent flowers pose a risk to bees, the authors measured the concentrations of several insecticides and fungicides shortly before and at several days after application in pumpkin. This is a relevant question and in principle their approach appears to be suitable to do this.

However, the methods section lacks important details about how the pesticides were applied. For example, there is no information about the formulations used, which is critical because co-formulants can significantly affect how well the active ingredients are absorbed, transported, and persist on the plants (as the authors acknowledge themselves in the discussion). Additionally, the timing of applications and whether applications were done before crop bloom or at night is missing, which is critical given their stated goal to measure potential risk from applications when flowers are closed or absent. Even the application rates are missing.

The authors introduce a new hazard quotient but its rationale is not sufficiently explained as well as it’s risk criterion. They claim that this is a more conservative measure than the conventional HQ but their results seem to suggest otherwise.

Although the experiment was designed to measure pesticide concentrations over time. I did not see any results on the pre-application measure. The proportion of positive samples is reported for post-application measures but I would appreciate a figure showing the temporal evolution of both the percentage of positive samples and their concentrations. I think in the discussion risk could be discussed more quantitively. Also, perhaps, what the high concentrations in leaves could mean for other insects (including pollinators, such as butterflies or even leaf-cutter bees) could be discussed.

In summary, I think this is a decent study on an important topic that needs some more details and explanations provided. After major revisions the article will be suitable for publication in PLOS ONE.

Additional comments can be found below:

Table 1: Indicate for LD50 values if they were determined over 24h, 48h, or 96h.

Methods: I miss information on the application of these substances. In what formulations were they applied? The co-formulants influence greatly the capacity of the substances to be absorbed / transported etc. It is also not clear to me whether applications were done on different days or whether some substances were applied in tank mixtures. What time of the day? The authors want to estimate the risk of exposure for bees when the application is done at a time when the flowers are closed. For this it would be good to know whether the flowers were actually closed. What were the application rates? This would be very important to know to judge how much of the substances actually remain available for bees and to compare to future results with potentially different application rates.

L212. Please add a reference. I don’t think there is a universal definition of the HQ. In fact, EFSA GD 2013 (https://www.efsa.europa.eu/sites/default/files/consultation/120920.pdf) states “A HQ is the ratio between the application rate in g/ha and the LD50oral or LD50contact in µg/bee, i.e. g/ha ÷ LD50”, which is different from your definition.

L212-225. The HQ (both EFSAs (former) HQ and yours) and the HWPPB are somewhat apples to oranges comparisons (which may be okay for simplicity if you are aware of the underlying assumptions). For the normal HQ it is at least apparent from the units that it is an apples-to-oranges comparison. For the HWPPB you seem to suggest that this would not be the case as you first convert toxicity into the same units as the chemical concentrations. However, you cannot really cancel out µg/kg nectar (or whatever matrix you look at) by µg/kg bee tissue.

L327: Is this an average over all insecticides? Or was in 75% of the cases at least one insecticide was found in leaves collected on the following day?

6. PLOS authors have the option to publish the peer review history of their article (what does this mean? ). If published, this will include your full peer review and any attached files.

**Do you want your identity to be public for this peer review?** For information about this choice, including consent withdrawal, please see our Privacy Policy .

Reviewer #1: No

Reviewer #2: No

Reviewer #3: No

---

## [Author Response · Author response to Decision Letter 0]

17 Dec 2024

We have reviewed the style requirements and formatted our manuscript to them to the best of our knowledge.

We used the base package in R to make figures. We have included our R code script file and associated data summary .csv file on our Open Science Framework project page, along with the raw data for this manuscript.

Added statement to the Study System section of the Methods. No permits were required. We obtained verbal permission from the land owners (who were also the growers) to access the field sites.

4. Thank you for stating the following financial disclosure: [This work was funded by United States Department of Agriculture through a National Institute of Food and Agriculture grant 2016-51181-25410 awarded to IK and KG.].

If this statement is not correct you must amend it as needed. Please include this amended Role of Funder statement in your cover letter; we will change the online submission form on your behalf.

The funder had no role in this study, so we have amended the Role of Funder statement as suggested and included the updated statement in the cover letter.

We have used the PACE tool to format our figure files to the PLOS requirements.

Reviewers' comments:

Reviewer's Responses to Questions

Comments to the Author

1. Is the manuscript technically sound, and do the data support the conclusions?

Reviewer #1: Yes

Reviewer #2: Yes

Reviewer #3: Yes

2. Has the statistical analysis been performed appropriately and rigorously?

Reviewer #1: Yes

Reviewer #2: Yes

Reviewer #3: Yes

3. Have the authors made all data underlying the findings in their manuscript fully available?

Reviewer #1: Yes

Reviewer #2: Yes

Reviewer #3: No

The raw data, as well as data summary file and R code we used to create the manuscript figures, is now publicly available through Open Science Framework (an online repository for scientific data) at: https://osf.io/mg5et/?view_only=527ab0b982b4423192136a86dbbe3e70

4. Is the manuscript presented in an intelligible fashion and written in standard English?

Reviewer #1: Yes

Reviewer #2: Yes

Reviewer #3: Yes

5. Review Comments to the Author

Below, we respond to all reviewers’ comments. Note, all line numbers below are in reference to the originally submitted manuscript, not to the revised version.

Reviewer #1:

The present manuscript was well written, its objectives and the rationale of the study were clearly stated, the methods were adequate. Experiments were conducted rigorously, with appropriate controls. The conclusions were drawn appropriately based on the data presented. However, some suggestions and comments were provided in the attached copy of the submitted text.

Reviewer #1’s in-text comments were copied from the manuscript document and addressed here.

L127. Was thiamethoxam used only on seeds? Could the insecticide have been incorporated into the plant after the seeds germinated? Why didn't you check the plant for the presence of thiamethoxam?

Yes, thiamethoxam was used only as a seed coating. Obregon et al. 2020 and Hung et al. 2024 both found low levels of seed coat thiamethoxam in squash pollen and nectar during flowering, and other studies have documented translocation of additional systemic insecticides to pollen and nectar in other crops. Since translocation of systemic insecticides and their impacts on pollinators was already known, we chose to focus instead on mid-season non-systemic foliar sprays applied while the plants were in bloom. We added a sentence to the study system description in the methods to explain.

L128. How often were foliar sprays applied?

Added reference to S1 Table where pesticide application dates are listed.

L130. Have there been losses of bees in the bee hives on the 4 farms that also keep or contract bees?

Good suggestion. Monitoring nearby honey bee hives for losses during the study period may have given us insight into whether our low detection of pesticides in foraging bees the day after spray was due to survivorship bias. We added a statement about this to the discussion in the section where we discussed survivorship bias. To the best of our knowledge, there are no published studies reporting bee die-offs from mid-season foliar-application of pesticides to pumpkin. For pre-bloom pesticide treatments, Willis Chan and Raine 2021 found that soil application (but not seed treatment or foliar spray) of neonicotinoids for squash plants reduced squash bee reproduction, but they could not distinguish if it was due to acute toxicity of pesticides or sublethal effects.

L149. Why not? Explain here. Isn't the day before the 7th?

Added explanation. We made the tradeoff of looking at fewer time points in order to evaluate risk to three bee taxa instead of honey bees only. Throughout the manuscript we refer to the day before spray as ‘Day -1,’ instead of the 7th day.

L202. For 8 days? I understand that you collected data 1d before and 1d after the chemical applications.

At L141 we state that pumpkin plant tissues were collected at Day -1, 1, 3, and 7. Foraging bees were only collected at Day -1 (before) and Day 1 after pesticide application (L147-149).

L316, 431. Note: The reviewer highlighted these lines, but no comments were attached.

L479. Squash bees may have tested positive because they feed mainly on squash pollen and nectar, which is in contact with the agrochemical studied.

Edited to clarify.

Reviewer #2:

The present work contains important information regarding the exposure of potential pollinators to pesticides, considering the application of these products when the flowers are no longer secreting nectar and/or have pollen available. The strength of the article is the fact that the authors collected data in the field, in a real situation of pesticide use and potential exposure of non-target insects. The weakness is related to the low number of repetitions, especially for the data obtained with the bees.

We agree and acknowledged in the manuscript (e.g. L400) that overall low sample size was a weakness of the study. However, we would like to note that the number of bee samples was actually greater than that of other tissues (number of total samples by tissue type: leaf = 32, pollen = 32, nectar = 28, bee = 69), although a much smaller portion of the samples (only two out of 69) tested positive for insecticides.

Here are some notes that I think the authors should pay more attention to:

Despite the importance of the information presented in the introduction section, this section is excessively long. On the other hand, more information about the translocation of pesticides in plants should be included in the text. Since it may be difficult to include this information in the introduction of the article, I believe it would be better presented in the discussion section.

The reviewer did not provide specific suggestions, so we shortened paragraph 3 of the introduction. The introduction focuses more on the effects of pesticides on pollinators than on general detection of pesticides in the crop plants, because that is where our paper makes the most novel contribution to the field – by documenting translocation of pesticides up a trophic level to their pollinators. We agree with the reviewer that an explanation on how pesticides translocate within plants is more useful in the discussion section, to help us understand patterns our detection in crop plant and bee tissues (L411 – 423, L424 – 440, L441 – 449). We added more references at L444 to better compare our pesticide concentrations in crop pollen and nectar to other studies.

In the materials and methods section, the bee species should be described rather than just mentioned generically.

The relevant life history traits of the bees that pollinate pumpkin in our region were described in the introduction on L78 – 87. We feel it is needed in the introduction instead of the methods to establish ways bees incur risk from foraging on pumpkin plants recently sprayed with foliar pesticides.

Regarding the materials and methods section, the description of the sample preparation for the pesticide quantification analyses could be better detailed.

Edited at L184 – 186 to better clarify that we did not conduct the pesticide quantification ourselves. We prepared the tissue samples as described at L174-185, then shipped them to a large commercial agricultural testing laboratory (SGS Chemical Company, Brookings, SD, USA).

The names of the pumpkin cultivars should be mentioned in the text, as well as the general growing conditions. Even more importantly, the environmental conditions should be specified: include data on temperature, sunlight, and precipitation.

Farmers all grew multiple cultivars per field (and did not provide us with information on which cultivars they grew) in order to appeal to their customers as these pumpkins were to be sold at small local farm stands.

We added information on the general growing conditions of the study region and the mean daily weather at L123. We also added a table to the supporting information with daily weather conditions.

What is the size of the cultivation areas? Could the bees have been contaminated in adjacent areas?

The reviewer raises a good point. It is possible that bees moved between adjacent fields since the area is intensively used for agriculture, each grower owned multiple fields, and bees can fly >1km in search of food if needed. However, it is not likely that these particular bees were contaminated in an adjacent field of a different crop that was not part of our study. The only two bee samples that tested positive for insecticide both consisted of male squash bees. Squash bees are unusual among bees in our study region in that they have a high degree of dietary specialization, and almost exclusively forage and nest on / near Cucurbita crop flowers. Since the pumpkin was blooming, it is unlikely that the bees would have left the field to forage on a less preferred food plant elsewhere during the study period. If bees had picked up pesticides from neighboring fields, it would have been evident in honey bees and bumble bees which have a much broader diet.

Some researchers have reported difficulties in detecting/quantifying pesticides in pollen samples. Is it possible to assert that the quantities observed in the pollen samples were not underestimated? I believe that a more detailed description of the detection method could better support the technique chosen for observing pesticides in pollen.

Edited to better support our choice of extraction method, and added a reference to Hall et al 2020, who reported rates of insecticide residue recovery of 89.4 – 101 % from pollen using the QuEChERS method (https://pmc.ncbi.nlm.nih.gov/articles/PMC7355641/). We used a common method for extracting residues of acidic pesticides from biological tissues. We were not aware that extra sample preparations needed to be made for pollen, as we did not find suggested modifications in the protocol or published literature for pollen. The QuEChERS method was validated for cucumber (among other crops), a close relative of pumpkin, and no mention was made of it having unsatisfactory performance for any of the pesticides we were working with. In a recent review of studies reporting pesticide residues in pollen and nectar from 1968 – 2018, 13 out of 25 articles used this same method. The 2 most recent papers that did not use that method for extracting pesticide residues from pollen created their own method that involved an extra extraction step using methanol, but they did not share their reasoning behind their methods.

If a larger number of bees had been collected, regression analyses could have been established to understand the exposure time of these insects after spraying.

We agree that a larger number of samples would have been useful in order to gain enough statistical power to do regression analysis. However, we chose to concentrate our bee sampling during the days immediately before and after spray (when exposure was most likely) in order to most clearly address our question of whether the pesticides translocation up the food chain from crop plant to bee. In doing so, we were able to screen more bee taxa to include an important solitary bee pollinator, but lost time resolution because we did not sample on days 3 and 7 post-treatment. Even though we collected twice as many samples of bees compared to each pumpkin tissue type (since less is known about bioaccumulation in pollinators than about translocation within the crop plant), our detection rate in bees was still very low.

Table 1 is not mentione

---

## [Decision Letter · Decision Letter 1]

25 Feb 2025

Short-term persistence of foliar insecticides and fungicides in pumpkin plants and their pollinators

PONE-D-24-42091R1

Dear Dr. Goodell,

We’re pleased to inform you that your manuscript has been judged scientifically suitable for publication and will be formally accepted for publication once it meets all outstanding technical requirements.

Kind regards,

Munir Ahmad, PhD

Academic Editor

PLOS ONE

Additional Editor Comments (optional):

The amendments listed by reviewer 2 do not need to be addressed.

Reviewers' comments:

Reviewer's Responses to Questions

**Comments to the Author**

1. If the authors have adequately addressed your comments raised in a previous round of review and you feel that this manuscript is now acceptable for publication, you may indicate that here to bypass the “Comments to the Author” section, enter your conflict of interest statement in the “Confidential to Editor” section, and submit your "Accept" recommendation.

Reviewer #1: All comments have been addressed

Reviewer #3: All comments have been addressed

2. Is the manuscript technically sound, and do the data support the conclusions?

Reviewer #1: Yes

Reviewer #3: Yes

3. Has the statistical analysis been performed appropriately and rigorously? 

Reviewer #1: Yes

Reviewer #3: Yes

4. Have the authors made all data underlying the findings in their manuscript fully available?

Reviewer #1: Yes

Reviewer #3: Yes

5. Is the manuscript presented in an intelligible fashion and written in standard English?

Reviewer #1: Yes

Reviewer #3: Yes

6. Review Comments to the Author

Reviewer #1: The manuscript has been appropriately revised. All my questions were answered by the authors. They also accepted my suggestions. I also consider that they properly answered the questions of other reviewers. The work is ready for publication.

Reviewer #3: The authors have adequately responded to my comments. If there is one point they could improve on I would it is still the explanation of what the real benefit of the newly proposed HQ is given that the risk criterion was not yet established, but generally I think they did a good job. I recommend accepting it for publication.

7. PLOS authors have the option to publish the peer review history of their article (what does this mean? ). If published, this will include your full peer review and any attached files.

**Do you want your identity to be public for this peer review?** For information about this choice, including consent withdrawal, please see our Privacy Policy .

Reviewer #1: No

Reviewer #3: No

---

## [Editor Report · Acceptance letter]

PONE-D-24-42091R1

PLOS ONE

Dear Dr. Goodell,

I'm pleased to inform you that your manuscript has been deemed suitable for publication in PLOS ONE. Congratulations! Your manuscript is now being handed over to our production team.

Kind regards,

on behalf of

Dr. PLOS Manuscript Reassignment

Staff Editor

PLOS ONE